# Effects of Magnetic Field Therapy and Massage on Upper Trapezius Muscle Tone, Craniovertebral Angle, and Scapular Index

**DOI:** 10.3390/bioengineering12090925

**Published:** 2025-08-28

**Authors:** Do-Youn Lee, Seong-Gil Kim

**Affiliations:** 1College of General Education, Kookmin University, Seoul 02707, Republic of Korea; triptoyoun@kookmin.ac.kr; 2Department of Physical Therapy, College of Health and Life Science, Korea National University of Transportation, Jeungpyeong-gun, Chungcheongbuk-do 27930, Republic of Korea

**Keywords:** magnetic fields, massage therapy, muscle tonus, posture, trapezius muscle, young adult

## Abstract

This study investigated the effects of magnetic field therapy and massage on upper trapezius muscle tone, craniovertebral angle (CVA), and scapular index in young adults. Thirty participants were randomly assigned to a magnetic field group or a massage group (n = 15 each), receiving interventions twice a week for two weeks with a one-week follow-up. Measurements were taken at baseline, post-intervention, and follow-up using MyotonPRO, lateral photographs, and anatomical distances. The magnetic group showed a greater reduction in muscle tone at post-intervention and follow-up (*p* = 0.015, partial η^2^ = 0.28, large effect) than the massage group. Elasticity decreased significantly in both groups, but follow-up values were lower in the magnetic group (*p* < 0.05, partial η^2^ = 0.25, medium effect). CVA improved in both groups, with a larger change in the magnetic group and sustained gains at follow-up (*p* < 0.001, partial η^2^ up to 0.43, large effect). The scapular index increased only in the magnetic group post-intervention (*p* = 0.013, partial η^2^ = 0.49, large effect) but returned to baseline at follow-up. Magnetic field therapy appears more effective than massage for improving muscle tone, posture, and scapular alignment in the short term, and may be a valuable option for clinical postural correction.

## 1. Introduction

Lifestyle habits in modern society, such as prolonged computer and smartphone use, have led to a prevalence of postural imbalances, including Forward Head Posture [1,2]. These structural problems cause chronic overload on specific muscles, and the upper trapezius is particularly susceptible to developing Myofascial Trigger Points, a key cause of neck pain and shoulder dysfunction [3]. The resulting misalignment can degrade objective indicators like the Craniovertebral Angle (CVA) and the Scapular Index, ultimately threatening overall musculoskeletal health [4].

The CVA and Scapular Index are critical clinical indicators for quantitatively assessing postural alignment [2]. The CVA measures the degree of forward head protrusion, where a smaller angle signifies a more severe posture. The Scapular Index reflects the degree of thoracic kyphosis, such as in rounded shoulders, thereby demonstrating postural imbalance quantitatively. Tension in the upper trapezius directly correlates with these indicators and has a decisive influence on cervical stability and the functional movement of the upper limbs [5,6].

In clinical practice, various interventions are used to address these issues. Traditional massage therapy, employing techniques like Myofascial Release and Trigger Point Therapy, is an established and effective treatment for alleviating tension, stiffness, and pain in the upper trapezius [7,8]. Research consistently reports that massage directly reduces muscle tone and stiffness, increases the pressure pain threshold, and significantly improves neck range of motion [9,10].

In parallel, Pulsed Electromagnetic Fields (PEMF) therapy has emerged as an innovative, non-invasive modality for managing musculoskeletal disorders [11]. PEMF functions by emitting low-frequency electromagnetic pulses deep into tissues to induce physiological changes at the cellular level [12]. Specifically, PEMF regulates cellular energy metabolic pathways, such as NAD+ signaling and oxidative phosphorylation [13], and increases levels of antioxidant proteins like Heat Shock Protein 70, which promotes muscle recovery and reduces inflammation [14]. Through these mechanisms, PEMF optimizes cellular energy metabolism and effectively controls inflammation and oxidative stress to promote fundamental tissue recovery. Clinically, PEMF has also been shown to significantly improve pain intensity and functional disability in various musculoskeletal pain conditions [15,16].

Both massage and PEMF therapy are expected to yield positive effects on upper trapezius dysfunction through their distinct mechanisms. However, there is a significant lack of research that directly compares the effects of massage, based on mechanical stimulation, and PEMF, based on biophysical stimulation, using objective indicators (muscle tone, CVA, Scapular Index) and tracks the persistence of these effects.

Therefore, this study aims to verify and compare the effects of magnetic therapy and massage on the muscle tone of the upper trapezius, the CVA, and the Scapular Index. The findings are intended to provide crucial scientific evidence for selecting the optimal treatment method for patients with postural imbalance, thereby informing clinical intervention choices.

## 2. Materials and Methods

### 2.1. Study Subjects

This study was conducted on a total of 30 university students (26 males, 4 females) attending S University in Chungcheongnam-do. The subjects’ mean age was 22.00 ± 2.18 years, mean height was 172.56 ± 4.94 cm, and mean weight was 75.67 ± 13.94 kg (Table 1).

The required sample size was calculated using G*Power 3.1.9.7 (Heinrich Heine University, Düsseldorf, Germany), with a significance level (α) of 0.05, power of 0.80, and an effect size of 0.6. The calculation indicated a minimum of 30 subjects was necessary. To account for potential dropouts, 33 subjects were initially recruited, but 3 dropped out during the study (Table 1).

The inclusion criteria for subjects were as follows: (1) adults aged 20 years or older, (2) no significant impairments in vision or somatosensation, (3) no pain in the abdominal or lumbar regions that could interfere with the intervention, (4) not taking medications affecting muscle tone and having a Body Mass Index (BMI) below 30 kg/m^2^, and (5) right-handed. All subjects were fully informed of the study’s purpose and procedures and provided voluntary written consent in accordance with the ethical standards of the Declaration of Helsinki before the experiment commenced. Table 1 shows the general characteristics of the participants.

### 2.2. Experimental Procedure

The subjects were randomly assigned to one of two intervention groups: the PEMF Group (n = 15) or the Massage Group (n = 15). Group assignment was performed according to a random number table generated using the RAND() function in Microsoft Excel (Microsoft Corp., Redmond, WA, USA). The intervention was administered twice a week for a total of two weeks (4 sessions). Assessments were conducted at three time points: baseline (pre), immediately after the final intervention (post), and one week after the conclusion of the intervention (follow-up).

The PEMF group received treatment on the upper trapezius muscle. The therapeutic head of a PEMF therapy device (G-500, Stratek, Anyang, Republic of Korea, Figure 1) was positioned over the area, delivering an intermittent stimulus at a frequency of 20 Hz (5 s of stimulation followed by 2 s of non-stimulation) for 20 min. This frequency was selected as it is known to have various biological effects, including tissue regeneration and bone formation, and is considered a key frequency in signal optimization studies [17]. The intensity of the electrical stimulation was adjusted to a level that did not cause pain or discomfort to the subject.

The massage group received massage therapy using the Graston Technique on the same area using stainless steel instruments (Graston Technique, LLC., Indianapolis, IN, USA). A skilled physical therapist continuously massaged the soft tissue for 20 min at a comfortable intensity that did not induce pain.

All interventions were performed by the same therapist in a quiet and controlled treatment room. To minimize mutual interference, the assessor and the practitioner were separated during the procedures. Furthermore, all outcomes were measured by a single assessor who was unaware of the participants’ group allocation, ensuring blinding of the assessment process.

### 2.3. Measurement Tools and Evaluation Methods

#### 2.3.1. Muscle Tone (MyotonPRO Measurement)

Muscle tone was assessed at the trigger point of the left (non-dominant) upper trapezius. A portable muscle tonometer, the MyotonPRO (Myoton AS, Tallinn, Estonia), was used for the measurement (Figure 2). Before measurement, the evaluator identified the most sensitive trigger point by palpation and marked it with a water-based pen. A 5-min rest period was provided to prevent tension responses from the measurement process itself.

Subjects sat in a chair looking forward, with both hands placed on their knees. The MyotonPRO probe was placed perpendicular to the marked trigger point, and a mechanical impulse of 0.4 N was applied 5 times. This process was repeated three times with a 15-s interval between each measurement, and the average value was used for analysis. The measured parameters were muscle tone (frequency, Hz), stiffness (N/m), and logarithmic decrement, which reflects elasticity and fatigue.

#### 2.3.2. Craniovertebral Angle (CVA)

The CVA was measured to quantitatively assess the degree of Forward Head Posture. Subjects were photographed from the side while sitting in a natural posture. The captured images were analyzed using Image J (Version 1.53, National Institutes of Health, Bethesda, MD, USA). The angle was measured between a line connecting the C7 spinous process and the tragus of the ear and a horizontal line (Figure 3). This angle serves as an indicator of the head’s anterior displacement; a smaller angle signifies a more severe forward head posture.

#### 2.3.3. Scapular Index

The Scapular Index was used to evaluate the anteroposterior imbalance between the thorax and the scapula. The Scapular Index was calculated using Equation (1):(1)$$Scapular\;Index\;(\%)=\frac{Anterior\;Distance}{Posterior\;Distance}\times 100$$

The Anterior Distance was measured from the center of the sternal notch to the medial aspect of the coracoid process. The Posterior Distance was measured from the postero-lateral tip of the acromion to the thoracic spinous process. The Scapular Index was calculated using the following formula. This index indirectly reflects the degree of round shoulders or kyphosis through the positional ratio of anterior and posterior structures, with lower values indicating more severe postural imbalance.

### 2.4. Statistical Analysis

All data analysis was performed using SPSS for Windows (version 26.0, IBM Corp., Armonk, NY, USA). First, a normality test was conducted to check the assumption of normality.

To analyze changes over the intervention time points (pre, post, follow-up), a one-way repeated measures ANOVA was used, with Fisher’s Least Significant Difference (LSD) test applied for post hoc analysis. To compare differences between groups, an independent *t*-test was conducted.

The statistical significance level for all tests was set at α = 0.05.

## 3. Results

Analysis of changes over time revealed that muscle tone showed a significant difference only in the Magnetic group (Table 2, Figure 2), with a decrease from 25.15 ± 0.95 Hz at pre-intervention to 24.33 ± 1.72 Hz at post-intervention (*p* = 0.015, partial η^2^ = 0.284, large effect) and to 24.17 ± 1.65 Hz at follow-up (*p* = 0.015, partial η^2^ = 0.284, large effect), whereas no significant changes were observed in the Massage group (24.62 ± 1.33 Hz → 24.30 ± 1.40 Hz → 24.80 ± 1.30 Hz). The detailed values of muscle tone are presented in Table 2.

For elasticity (Table 2, Figure 2), significant changes were observed in both groups. The Massage group improved from 1.04 ± 0.07 at pre-intervention to 1.02 ± 0.03 at post-intervention (*p* = 0.014, partial η^2^ = 0.252, medium effect), whereas the Magnetic group improved from 1.04 ± 0.06 at pre-intervention to 1.00 ± 0.06 at follow-up (*p* = 0.027, partial η^2^ = 0.252, medium effect). The between-group comparison at follow-up showed significantly lower elasticity in the Magnetic group (1.00 ± 0.06) compared to the Massage group (1.03 ± 0.03, *p* = 0.030, partial η^2^ = 0.252, medium effect). These results are also summarized in Table 2.

The CVA (Table 3, Figure 3) increased significantly in both groups. The Massage group improved from 49.66 ± 1.05° at pre-intervention to 50.66 ± 1.05° at post-intervention (*p* < 0.001, partial η^2^ = 0.43, large effect) and to 50.94 ± 0.47° at follow-up (*p* < 0.001). The Magnetic group improved from 50.30 ± 1.38° at pre-intervention to 50.73 ± 1.20° at post-intervention (*p* < 0.001, partial η^2^ = 0.33, large effect) and to 51.13 ± 0.91° at follow-up (*p* < 0.001).

The Scapular Index (Table 3) exhibited a significant difference only in the Magnetic group, increasing from 90.99 ± 2.53% at pre-intervention to 92.41 ± 2.87% at post-intervention (*p* = 0.013, partial η^2^ = 0.489, large effect), but returning to 89.96 ± 4.76% at follow-up. The detailed outcomes of CVA and Scapular Index are presented in Table 3.

## 4. Discussion

This study analyzed changes in the upper trapezius muscle tone, CVA, and Scapular Index in adults in their 20s following the application of massage and magnetic therapy. Assessments were conducted at three time points: pre-intervention, post-intervention, and one-week follow-up.

Among the muscle tone-related indicators, muscle tone (frequency) significantly decreased in the Magnetic group from pre (25.23 ± 0.95 Hz) to post (24.33 ± 1.72 Hz, *p* = 0.015, partial η^2^ = 0.28, large effect) and follow-up (24.17 ± 1.65 Hz, *p* = 0.015, partial η^2^ = 0.28, large effect), whereas no significant changes were observed in the Massage group. This sustained reduction indicates that magnetic therapy may produce long-lasting neuromuscular relaxation effects by influencing both contractile and passive tissue properties, as supported by recent PEMF-based muscle recovery trials [13,14,18,19].

Elasticity (logarithmic decrement) also showed significant changes in both groups, with the Massage group improving from pre (1.04 ± 0.07) to post (1.02 ± 0.03, *p* = 0.014, partial η^2^ = 0.25, medium effect), and the Magnetic group improving from pre (1.04 ± 0.06) to follow-up (1.00 ± 0.06, *p* = 0.027, partial η^2^ = 0.25, medium effect). Elasticity, represented by the logarithmic decrement of muscle oscillation, indicates that a lower value corresponds to higher restorative capacity and lower fatigue [13,14,19,20,21,22,23]. Notably, between-group analysis at follow-up showed significantly lower elasticity in the Magnetic group compared to the Massage group (*p* = 0.030, partial η^2^ = 0.25, medium effect). Combined with the finding that muscle tone decreased significantly only in the Magnetic group, these results suggest that magnetic therapy is more effective than massage in improving muscle tone and fatigue.

This difference can be attributed to the direct influence of magnetic therapy on neuromuscular function and metabolic responses at the tissue level. Specifically, PEMF may alter cell membrane potential, regulate intracellular ion balance, and modulate calcium influx, thereby affecting muscle contractile proteins [11,12,13,14,24,25]. Furthermore, magnetic fields induce vasodilation and improve blood flow, which facilitates the removal of metabolic byproducts and enhances oxygen supply, ultimately contributing to muscle fatigue recovery and reduction of tissue inflammation [11,12,26,27]. In contrast, while massage induces temporary, localized increases in blood flow and myofascial release through mechanical soft tissue stimulation, its effects may be relatively superficial and transient [28]. Therefore, while it can provide short-term relaxation, its sustainability at the follow-up point appears to be less than that of magnetic therapy. In conclusion, magnetic therapy acts on the nervous, vascular, and muscular systems in a composite manner, inducing physiological changes in deeper tissues and thus proving to be an effective intervention for alleviating muscle tone and enhancing fatigue recovery.

The CVA increased significantly from pre to post in both groups (Massage: 49.66 ± 1.05° → 50.66 ± 1.05°, *p <* 0.001, partial *η^2^* = 0.43, large effect; Magnetic: 50.30 ± 1.38° → 50.73 ± 1.20°, *p* < 0.001, partial η^2^ = 0.33, large effect), indicating an improvement in forward head posture. The improvement in the Massage group was maintained through the follow-up (51.13 ± 0.91°, *p* < 0.001), whereas the Magnetic group not only improved at post-test but also showed a further significant increase at the follow-up assessment. This result suggests that magnetic therapy is more effective for improving CVA, which aligns with previous research indicating that magnetic fields affect deep tissues to relieve tension in the muscles around the cervical spine and improve postural alignment [29,30,31]. Conversely, massage provides direct stimulation to superficial muscles, yielding temporary relaxation, but may have less sustained effectiveness in long-term postural alignment compared to magnetic therapy.

The Scapular Index showed a significant increase only in the Magnetic group at the post-intervention time point (90.99 ± 2.53% → 92.41 ± 2.87%, *p* = 0.013, partial η^2^ = 0.49, large effect), indicating an improvement in round shoulder posture. However, this effect was not sustained at the follow-up, as the index decreased again. The Scapular Index reflects the balance between anterior and posterior thoracic structures; a higher value indicates a more aligned posture with the scapulae positioned more posteriorly and reduced anterior protrusion. The temporary improvement in round shoulder posture by magnetic therapy may be due to its physiological induction of relaxation in the trapezius and surrounding scapular muscles, which assists spinal alignment. However, without accompanying long-term muscle re-education, proper gait, and postural habits, the effect is limited at the follow-up stage [32,33].

In summary, while both massage and magnetic therapy were effective in reducing upper trapezius fatigue and improving forward head posture (CVA increase), the Magnetic group demonstrated greater and more sustained benefits, with large effect sizes (partial η^2^ ≥ 0.25) observed in muscle tone, elasticity, and postural measures such as the Scapular Index. These findings suggest that magnetic therapy exerts deeper and more comprehensive physiological effects than physical massage by modulating neuromuscular excitability, enhancing microcirculation, and promoting autonomic nervous system stabilization and tissue recovery. The results of this study provide evidence supporting the clinical integration of magnetic therapy for both preventive and rehabilitative purposes in postural correction and muscle fatigue reduction. Furthermore, the combination of magnetic therapy with conventional manual therapy or exercise programs could potentially amplify these benefits, which should be explored in future trials.

The limitations of this study include the small sample size, short intervention duration, and inclusion of only healthy young adults, which may limit generalizability to older populations or clinical groups. Future research should recruit larger and more diverse cohorts, extend the follow-up period to evaluate long-term retention of effects, and investigate combined interventions that incorporate motor control or posture re-education strategies to enhance lasting outcomes.

## 5. Conclusions

This study compared the effects of massage and magnetic therapy on the muscle tone of the upper trapezius, CVA, and the Scapular Index in healthy young adults. Both interventions were effective in improving forward head posture and reducing fatigue. However, magnetic therapy demonstrated greater and more sustained improvements, with large effect sizes (partial η^2^ ≥ 0.25) observed in muscle tone, elasticity, and postural outcomes.

These findings suggest that magnetic therapy can induce deeper and longer-lasting physiological changes by acting synergistically on muscles, nerves, and blood flow, leading to enhanced postural correction and muscle fatigue recovery. This supports its potential integration into preventive and rehabilitative physical therapy programs, particularly for conditions involving muscle overuse or postural imbalance.

Future studies should recruit larger and more diverse populations, extend follow-up periods, and explore combined interventions with motor control or posture re-education strategies to maximize and sustain therapeutic benefits.

## Figures and Tables

**Figure 1 bioengineering-12-00925-f001:**
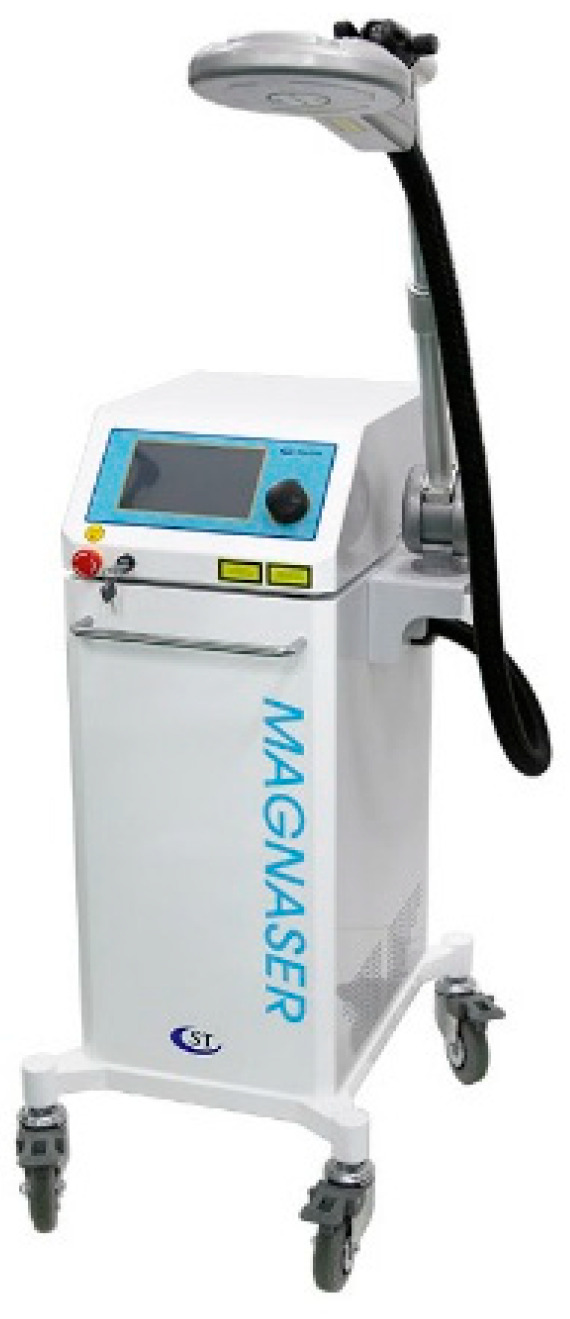
Magnetic field therapy device.

**Figure 2 bioengineering-12-00925-f002:**
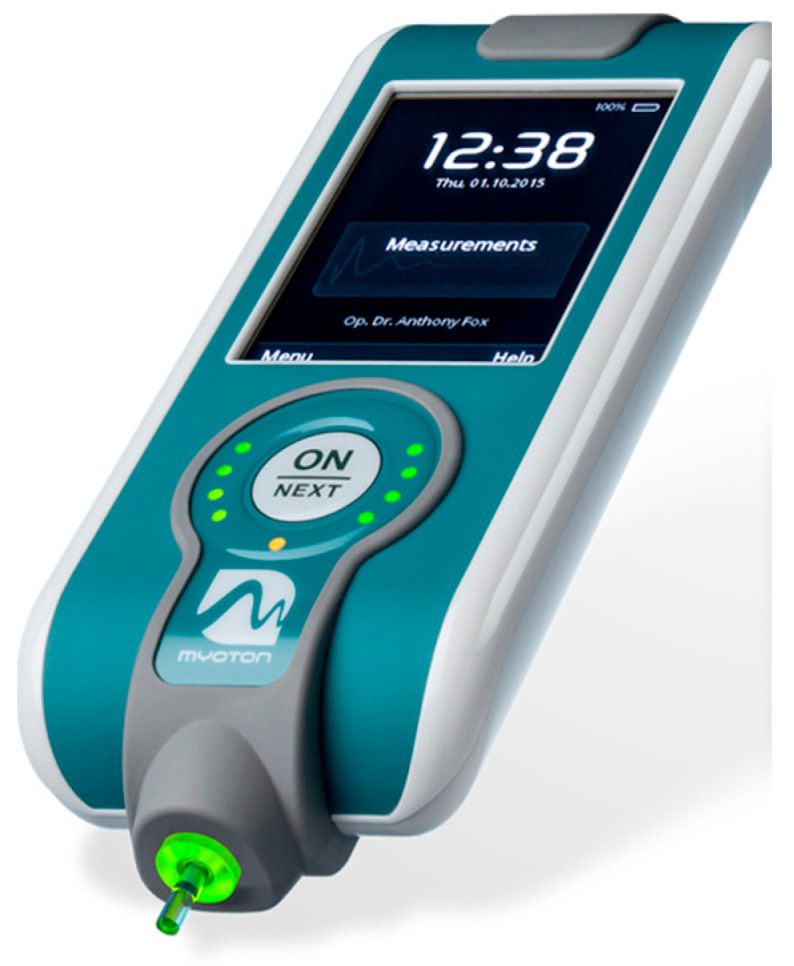
MyotonPRO.

**Figure 3 bioengineering-12-00925-f003:**
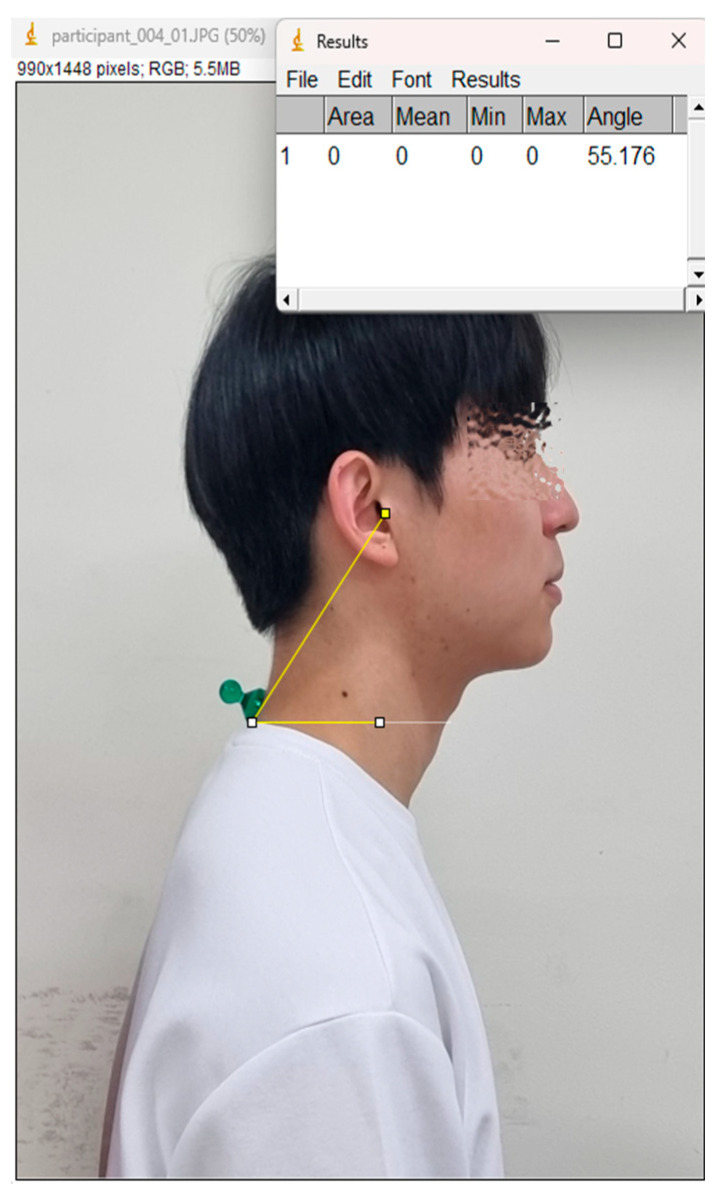
Craniovertebral Angle (CVA).

**Table 1 bioengineering-12-00925-t001:** General subject characteristics (n= 30).

Variable	Mean ± SD
Age (year)	22.00 ± 2.18
Height (cm)	172.56 ± 4.94
Weight (kg)	75.67 ± 13.94

SD, standard deviation.

**Table 2 bioengineering-12-00925-t002:** Changes in Muscle Tone, Stiffness, and Elasticity of the Upper Trapezius During the Intervention Period.

Variable	Group	Pre	Post	Follow-Up	F	*p*	Partial η^2^ (Effect Size)
Muscle tone (Hz)	Massage	24.62 ± 1.33	24.30 ± 1.40	24.80 ± 1.30	1.672	0.198	0.060
	Magnetic	25.15 ± 0.95 ^§‖^	24.33 ± 1.72 ^‡^	24.17 ± 1.65 ^‡^	4.961	0.015 * Pre > Post = Followup	0.284
	*p*	0.101	0.938	0.128			
Stiffness (N/m)	Massage	398.85 ± 10.25	396.07 ± 9.06	396.89 ± 13.38	0.385	0.682	0.015
	Magnetic	394.89 ± 4.47	391.85 ± 12.94	389.72 ± 15.61	2.510	0.102	0.167
	*p*	0.074	0.171	0.076			
Elasticity (Log decrement)	Massage	1.04 ± 0.07	1.02 ± 0.03 ^‡^	1.04 ± 0.08	5.082	0.014 * Pre = Followup > Post	0.289
	Magnetic	1.04 ± 0.06 ^‖^	1.04 ± 0.07	1.00 ± 0.06 ^†‡^	4.202	0.027 * Pre = Post > Followup	0.252
	*p*	0.921	0.090	0.030 *			

(Mean ± SD), * *p* < 0.05, ^†^ Statistically different between groups, ^‡^ Statistically different from pre, ^§^ Statistically different from post, ^‖^ Statistically different from Follow-up. Between-group *p*-values at each time point are shown in the second row for each variable.

**Table 3 bioengineering-12-00925-t003:** Changes in Craniovertebral Angle (CVA) and Scapular Index During the Intervention Period.

Variable	Group	Pre	Post	Follow-Up	F	*p*	Partial η^2^ (Effect Size)
craniovertebral angle (degree)	Massage	49.66 ± 1.05 ^§‖^	50.66 ± 1.05 ^‡^	50.94 ± 0.47 ^‡^	19.555	0.000 * Pre < Post = Followup	0.429
	Magnetic	50.30 ± 1.38 ^§‖^	50.73 ± 1.20 ^‡‖^	51.13 ± 0.91 ^‡§^	12.712	0.000 * Pre < Post < Followup	0.328
	*p*	0.059	0.810	0.351			
Scapular Index (%)	Massage	92.05 ± 1.80	92.63 ± 3.20	91.20 ± 2.24	1.475	0.251	0.118
	Magnetic	90.99 ± 2.52 ^§^	92.41 ± 2.87 ^‡‖^	89.96 ± 4.76 ^§^	6.225	0.013 * Pre = Followup < Post	0.489
	*p*	0.140	0.793	0.315			

(Mean ± SD), * *p* < 0.05, ^‡^ Statistically different from pre, ^§^ Statistically different from post, ^‖^ Statistically different from Follow-up. Between-group *p*-values at each time point are shown in the second row for each variable.

## Data Availability

Data supporting the reported results are not publicly available due to privacy/ethical restrictions.

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
