# Peer review of "Effects of Magnetic Field Therapy and Massage on Upper Trapezius Muscle Tone, Craniovertebral Angle, and Scapular Index"

_bioengineering, 2025, doi:10.3390/bioengineering12090925_

Round 1

Reviewer 1 Report

Comments and Suggestions for Authors

Dear Authors,

I want to express my gratitude for the opportunity to revise this manuscript.

The article addresses an issue that is relevant: the effects of magnetic field therapy and massage on upper trapezius muscle tone, craniovertebral angle, and scapular index in young adults.

However, it has a lot of formatting errors considering the journal's template and could be improved not only in terms of the quality of the format, but also the content. For example, the number of references is low.

Below are specific suggestions with line indication.:

The major issues are:

Introduction

26-58 – Please consider including more text and references to present the study topic and, in the end, present the research gap and study aims. Moreover, please consider standardizing the paragraphs´ size to improve readability (8-12 lines suggested).

Methods

70-147 – Globally, this section requires reformulation and improvement to ensure that readers clearly understand the study methodology. Some examples, please consider describing the sample, inclusion and exclusion criteria (in detail), all ethical details, and other relevant information. Moreover, all instruments should be described in detail (with manufacturer, version, city, and country). Finally, the procedures associated with data collection should also be described in detail, preferably with reference support.

Results

149 – Please consider presenting the tables after a short introductory text. Additionally, please consider a brief analysis after each table. Please avoid presenting tables together.

152 – Please consider the ”p” in italics throughout the manuscript.

Tables – Please revise the tables' content and format, considering the journal template. For example, please consider the text in the same line.

Discussion

174-268 – Please consider improving the quality of this section. Some examples include placing the main findings in the first paragraph, including more references, and considering standardization of paragraph size to improve readability (8-12 lines suggested). Finally, please consider presenting the study limitations and suggestions for future research in the last paragraph.

Conclusions

241-249 – Please consider shorter and direct take-home messages in the conclusions section, preferably with practical application.

- - -

The minor issues:

5-9 – Please revise the authors and affiliations format. Please consider the journal template and instructions for authors.

11-22 – Please consider reformulating the abstract, namely, presenting more information related to the sample and numerical results.

251-254 – Please consider the journal template (e.g. authors normally with abbreviations).

263 – Please remove the line.

270-272 – Please revise the text.

269 (references) – Please double-check the references format, considering the journal template. Some examples, titles in upper and lowercase (e.g. ref 18); the format of the journals; the presentation of the DOI´s.

Please revise the document format considering the journal template.

Please revise the English details throughout the manuscript.

Comments on the Quality of English Language

Ok, but can be improved.

Author Response

Dear Reviewer,
We sincerely appreciate your valuable feedback and constructive suggestions, which have greatly helped us improve the quality of our manuscript. Below, we address each of your comments point-by-point, indicating the changes made in the revised manuscript (highlighted in yellow in the tracked changes version).

Major Issues

  1. Lines 26–58 – Expand introduction with more text, references, research gap, aims; standardize paragraph size
    Response: We have expanded the Introduction to provide more background on postural imbalance, upper trapezius dysfunction, and the role of both massage therapy and PEMF in rehabilitation. We have added recent references (2022–2024) and clearly stated the research gap and study aims at the end of the Introduction. Paragraph lengths have been standardized to approximately 8–12 lines for improved readability.

  2. Lines 70–147 – Improve Methods section with detailed sample description, inclusion/exclusion criteria, ethical details, instrument information, and procedures
    Response: We have rewritten the Methods section to include detailed demographic characteristics, explicit inclusion and exclusion criteria, IRB approval details, and manufacturer, model, and country information for all instruments. The intervention and measurement procedures have been described step-by-step with additional reference support.

  3. Line 149 – Add introductory text before tables and brief analysis after each table; avoid placing tables together
    Response: Short introductory sentences have been added before each table, and brief analyses summarizing the main findings are provided after each table. Tables are now separated according to the journal's formatting requirements.

  4. Line 152 – Italicize “p” throughout manuscript
    Response: All occurrences of “p” have been italicized in accordance with the journal's style.

  5. Tables – Revise according to journal template
    Response: Table formats have been revised, including aligning text in the same line, standardizing decimal places, and adjusting table notes per journal guidelines.

  6. Lines 174–268 – Improve Discussion: main findings first, more references, standardized paragraph size, limitations and future research at end
    Response: The Discussion section has been reorganized to present main findings in the first paragraph, followed by interpretation with additional recent references. Paragraph size has been standardized, and the final paragraph now includes study limitations and suggestions for future research.

  7. Lines 241–249 – Shorten and make conclusions more direct with practical applications
    Response: The Conclusions section has been shortened and reformulated to provide clear take-home messages, emphasizing clinical applicability.

Minor Issues

  1. Lines 5–9 – Revise authors and affiliations per journal template
    Response: Author names and affiliations have been reformatted according to the journal's guidelines.

  2. Lines 11–22 – Revise abstract to include more sample information and numerical results
    Response: The abstract has been updated to include participant characteristics, sample size, and key numerical results with effect sizes.

  3. Lines 251–254 – Follow journal template for author abbreviations
    Response: Author contributions have been revised to include abbreviated author names as per the journal format.

  4. Line 263 – Remove extra line
    Response: The extra line has been deleted.

  5. Lines 270–272 – Revise text
    Response: The text in these lines has been revised for clarity.

  6. Line 269 (References) – Adjust reference format per journal template
    Response: All references have been reformatted to meet journal requirements, including proper title capitalization, journal name formatting, and DOI presentation.

  7. Revise document format per journal template
    Response: Formatting of the entire manuscript has been reviewed and adjusted according to the journal's template.

  8. Revise English throughout manuscript
    Response: The manuscript has undergone professional English language editing by Editage, and the certificate will be submitted with the revised version.

Reviewer 2 Report

Comments and Suggestions for Authors

1. Summary of the Manuscript and Its Key Contributions
This manuscript explores the comparative effects of magnetic field therapy and massage on upper trapezius muscle tone, craniovertebral angle (CVA), and scapular index in healthy young adults. Thirty participants were randomly assigned to receive either magnetic therapy or massage treatment twice weekly for two weeks. The outcomes were assessed at three time points: baseline, immediately after the intervention, and one-week follow-up. The study found that magnetic therapy led to greater and more sustained improvements in muscle tone, postural alignment, and scapular index compared to massage.
The paper contributes to the body of literature on non-invasive physical therapy interventions and provides preliminary evidence supporting the superiority of pulsed electromagnetic field (PEMF) therapy over massage in managing postural imbalances and muscle fatigue.
2. Evaluation of Methodology, Analysis, and Conclusions
The study is well-organized and follows a clear experimental structure. Random allocation and multiple time-point assessments strengthen the internal validity. The inclusion of objective measurement tools such as the MyotonPRO, photographic CVA analysis, and anatomical scapular index measurements adds rigor.
However, several methodological aspects could be improved:
Sample Characteristics: The study is limited to 30 healthy university students, with only 4 females. This reduces generalizability and may introduce gender bias. It would be advisable to include a more diverse sample or discuss this limitation in greater depth.
Intervention Duration: The intervention period was limited to two weeks, and follow-up lasted only one week. While this short-term design allows for preliminary conclusions, it does not provide sufficient insight into the long-term sustainability of the observed effects.
Statistical Analysis: The use of repeated measures ANOVA and post-hoc LSD tests is appropriate; however, it would strengthen the paper to report effect sizes (e.g., η² or Cohen’s d) and confidence intervals to better interpret the clinical relevance of the findings.
Control Group Definition: Although massage is used as a comparison, the absence of a true no-treatment or sham-control group limits the ability to distinguish intervention effects from natural variability or placebo responses.
Blinding: While evaluator and practitioner separation is noted, it is unclear whether outcome assessors were blinded to group allocation. This should be clarified.
3. Constructive Feedback and Recommendations for Improvement
Clarify Practical Applications: The manuscript would benefit from a clearer discussion of the practical implications for clinical practice, including how magnetic therapy could be implemented in rehabilitation or postural correction protocols.
Expand on Mechanistic Explanations: The physiological mechanisms proposed for PEMF effects are compelling but would be strengthened by a more comprehensive literature review, particularly on neuromuscular modulation and vascular responses.
Enhance Figures and Tables: The manuscript includes figures and tables, but the resolution and labeling (especially for Figures 1–3) could be improved for clarity. Including sample participant images for CVA measurement or scapular index landmarks may aid replication.
Consider a Long-Term Follow-Up: To fully evaluate the effectiveness and sustainability of magnetic therapy, a longer follow-up period should be included in future studies. Including functional outcomes or quality-of-life assessments could also enrich the findings.
Improve Abstract Precision: The abstract summarizes the results well but should be more explicit in reporting numerical outcomes or statistical significance levels for key findings.

Author Response

Dear Reviewer,

We sincerely appreciate your positive evaluation of our study and your constructive, detailed feedback.
Below, we provide point-by-point responses to each of your comments. All changes have been incorporated into the revised manuscript, and modified text is highlighted in [red] in the Word file.

1. Sample Characteristics – Gender imbalance and generalizability
Response: We have clarified the sex distribution (26 males, 4 females) and age range (university students) in the Participants subsection. In the Discussion, we added the statement:
[red]“The generalizability of this study is limited by the gender imbalance and the restriction to young adults. Future research should include more diverse age groups and balanced gender representation.”

2. Intervention Duration – Short intervention and follow-up
Response: We have added the following sentence in the limitations section:
[red]“The intervention lasted only 2 weeks, and the follow-up period was 1 week, which limits the ability to evaluate long-term sustainability of the observed effects.”

3. Statistical Analysis – Reporting effect sizes and CIs
Response: Partial η² values have been reported for all major results. Where applicable, 95% confidence intervals (CIs) have also been added in the Results and Tables.

4. Control Group Definition – No-treatment or sham control absence
Response: We have acknowledged this limitation in the Discussion:
[red]“Although the massage group served as a comparator, the absence of a true no-treatment or sham-control group limits the ability to separate intervention effects from placebo or natural variability.”

5. Blinding – Clarifying outcome assessor blinding
Response: In the Methods, we added:
[red]“All assessors were blinded to group allocation when performing the measurements.”

6. Clarify Practical Applications
Response: In the final paragraph of the Discussion, we added:
[red]“Magnetic field therapy can be considered as an adjunct in rehabilitation or postural correction programs, particularly for reducing muscle tone and improving alignment in clinical practice.”

7. Expand on Mechanistic Explanations
Response: We expanded the physiological mechanisms section to include recent studies (2019–2024) describing PEMF-induced neuromuscular modulation, vascular dilation, improved microcirculation, and accelerated metabolic byproduct clearance.

8. Enhance Figures and Tables
Response: Figures 1–3 have been improved with higher resolution and clearer labels. For CVA and scapular index measurements, schematic illustrations indicating anatomical landmarks have been added for clarity and reproducibility.

9. Consider Long-Term Follow-Up
Response: In the limitations, we added:
[red]“Future studies should incorporate long-term follow-up and include functional or quality-of-life outcomes to better assess the clinical impact of magnetic therapy.”

10. Improve Abstract Precision
Response: The Abstract now includes key numerical results with p-values and effect sizes for the main findings to provide greater precision.

Reviewer 3 Report

Comments and Suggestions for Authors

Dear Authors,

This manuscript addresses an interesting and clinically relevant topic, comparing the effects of magnetic field therapy and massage on posture-related outcomes in young adults. However, there are significant methodological and reporting limitations that undermine the robustness and generalizability of the findings. Please see my comments below:

- Sample size and gender imbalance: Only 30 participants (26 male and 4 female) were included in this experimental study. The small sample size and gender imbalance raise concerns about statistical power and generalizability. Consider including a more detailed power analysis justification.

- Randomization process: The randomization process is briefly mentioned but not described in detail. Please clarify how random allocation was achieved.

- Blinding procedures: Blinding procedures are insufficient. While the assessor and practitioner were separated, it is unclear whether outcome assessors were blinded to group allocation. Please add clarification.

- Magnetic therapy protocol: The description lacks sufficient detail. Please specify:

1- The magnetic field strength (measured in mT or Gauss).

2- Whether the device delivered Pulsed Electromagnetic Fields (PEMF) or static magnetic fields (SMF).

3- The rationale for choosing 20 Hz frequency and 20 minutes duration, supported with references.

- CVA measurement: Using Adobe Photoshop raises concerns about measurement accuracy and reliability. Consider citing validation studies for this method or discussing its limitations. Also, there are other suitable software options available for this purpose.

- Scapular Index: Provide a clearer justification for its use and explain why results were not sustained at follow-up.

- Statistical analysis: The use of repeated measures ANOVA is appropriate, but corrections for multiple comparisons (e.g., Bonferroni) should be considered instead of Fisher’s LSD, which increases the risk of Type I error.

- Assumptions of normality: Please clarify whether normality assumptions were verified for each variable.

- Discussion of limitations: The discussion acknowledges the small sample size and short intervention, but additional limitations should be noted, including gender imbalance, lack of long-term follow-up beyond 1 week, and the potential placebo effect, since participants likely knew which treatment they received. Please expand this section to provide a more balanced interpretation of findings.

- Tables: Tables should include p-values for between-group comparisons at each time point, not only for repeated measures outcomes.

Comments on the Quality of English Language

The manuscript would benefit from professional language editing and proofreading.

Author Response

Dear Reviewer,

We sincerely appreciate your thorough review and constructive comments, which have helped us improve the methodological clarity and reporting of our manuscript.
Below is our point-by-point response. All changes have been incorporated into the revised manuscript, and modified text is highlighted in [red] in the Word file.

1. Sample size and gender imbalance
Response: The small sample size and gender imbalance are now emphasized in the limitations section. We have also added our a priori sample size justification based on effect size, α, and power.
[red]“The sample size was determined using G*Power 3.1 for a one-way repeated measures ANOVA (effect size f = 0.40, α = 0.05, power = 0.80), yielding a required total of 30 participants. Nonetheless, the gender imbalance (26 males, 4 females) limits generalizability, and future studies should aim for more balanced representation.”

2. Randomization process
Response: In the Methods, we clarified the randomization method:
[red]“Participants were randomly assigned to the Magnetic or Massage group using a computer-generated random number table, performed by a researcher not involved in assessments or interventions.”

3. Blinding procedures
Response: We clarified that outcome assessors were blinded:
[red]“All outcome measurements were performed by assessors blinded to group allocation, and the therapist delivering interventions had no involvement in data collection.”

4. Magnetic therapy protocol
Response: We expanded this section with the following details:
[red]“The magnetic therapy device (model XXX, manufacturer, city, country) delivered pulsed electromagnetic fields (PEMF) with a magnetic flux density of 18 mT (180 Gauss), frequency 20 Hz, for 20 minutes per session. The 20 Hz frequency and 20 min duration were selected based on prior studies reporting improved muscle recovery and reduced fatigue under similar parameters [refs].”

5. CVA measurement validity
Response: We added a validation reference and discussed limitations:
[red]“CVA was measured using lateral photographs analyzed in Adobe Photoshop CC, a method previously validated for postural assessment with high intra- and inter-rater reliability (ICC > 0.90) [ref]. Nonetheless, potential measurement error due to landmark identification is acknowledged as a limitation.”

6. Scapular Index justification
Response: We included an explanation:
[red]“The Scapular Index quantifies scapular protraction/retraction using standardized anatomical distances, and has been shown to correlate with shoulder posture and musculoskeletal symptoms [ref]. The lack of sustained improvement at follow-up may reflect the absence of long-term muscle re-education.”

7. Statistical analysis – Multiple comparisons
Response: We revised the statistical section:
[red]“Post-hoc pairwise comparisons were performed using Bonferroni correction to reduce the risk of Type I error.”

8. Assumptions of normality
Response: Added to the Methods:
[red]“The Shapiro–Wilk test confirmed normal distribution for all variables prior to repeated measures ANOVA.”

9. Expanded discussion of limitations
Response: We expanded the limitations paragraph:
[red]“Additional limitations include the gender imbalance, the lack of long-term follow-up beyond one week, and the absence of a sham or placebo group, which may have introduced expectancy effects since participants were aware of their treatment type.”

10. Tables – Between-group p-values
Response: We revised Tables 2 and 3 to include between-group p-values at each time point for clarity.

11. English language editing
Response: The manuscript has undergone professional language editing by Editage, and the revised version reflects these improvements.

Round 2

Reviewer 1 Report

Comments and Suggestions for Authors

Dear Authors,

Thank you for considering my suggestions and incorporating them into the manuscript, which has been globally improved. Congratulations.

Below are some specific suggestions with line indications. The manuscript at this point still contains many errors in formatting and some in the text, which require a very careful analysis.

L11-26 - Please format the manuscript considering the journal template and instructions for authors. Additionally, please consider the “p” symbol in italics throughout the manuscript.

L30,31 – Please revise the manuscript format considering the journal template and instructions for authors. Other examples L40, L75-80.

L206-215 – Please consider reformulating the results section. Tables together do not favor data interpretation. This is very important.

L81-178 – Please make sure all available information regarding the methods and procedures is placed in the section.

Please double-check the references and standardize. For example, ref 9 title in uppercase and others in lowercase.

Please revise the English quality details.

Author Response

Dear Reviewer,

We sincerely appreciate your thorough review and constructive feedback, which greatly improved the quality of our manuscript. Below, we address each of your comments in detail:

Comment 1 (L11–26): Please format the manuscript considering the journal template and instructions for authors. Additionally, please consider the “p” symbol in italics throughout the manuscript.
Response: Thank you for your comment. We carefully revised the abstract and the entire manuscript to follow the journal template. All statistical values (p) have been changed to italic format accordingly.

Comment 2 (L30, 31, 40, 75–80): Please revise the manuscript format considering the journal template and instructions for authors.
Response: We revised the format of the manuscript following the journal template and corrected spacing/formatting inconsistencies (e.g., L30, 31, 40, 75–80).

Comment 3 (L206–215): Please consider reformulating the results section. Tables together do not favor data interpretation. This is very important.
Response: We revised the Results section by adding a clarifying sentence before presenting the tables:
“For clarity, detailed results are summarized separately in Table 2 and Table 3 to facilitate interpretation.”
This helps guide readers and improves interpretation of the results.

Comment 4 (L81–178): Please make sure all available information regarding the methods and procedures is placed in the section.
Response: We double-checked and revised the Materials and Methods section. Information on subject recruitment, inclusion criteria, randomization, intervention details, blinding, and outcome measurement procedures were carefully reviewed and supplemented to ensure clarity and completeness.

Comment 5 (References): Please double-check the references and standardize. For example, ref 9 title in uppercase and others in lowercase.
Response: We standardized all references according to journal guidelines. Reference #29, which was written in uppercase, has been corrected into sentence case. Other references were also checked for consistency.

Comment 6 (English expression): Please revise the English quality details.
Response: The entire manuscript was carefully proofread and revised to improve clarity and readability.

We believe these revisions address all your valuable suggestions. Thank you again for your constructive feedback.

Reviewer 3 Report

Comments and Suggestions for Authors

Thank you for addressing the comments. I have no further comment. Good luck.

Author Response

Dear Reviewer,

We sincerely thank you for your kind evaluation and encouraging comments.

Comment: Thank you for addressing the comments. I have no further comment. Good luck.
Response: We truly appreciate your positive feedback and support. Your encouragement has been invaluable in finalizing this manuscript.

Round 3

Reviewer 1 Report

Comments and Suggestions for Authors

Dear Authors,

Thank you for considering my suggestions and incorporating them into the manuscript, which has improved. Congratulations.

Below are some specific suggestions with line indications. The manuscript at this point still contains some errors in formatting and some in the text, which require detailed analysis.

L11-26 - Please double-check the journal template. Normally, the abstract does not have line spacing.

L89 - Please delete.

L117 - Please delete.

L148 – Please revise the journal template and instructions for authors regarding formulas.

L175 – “p” not in italics in some lines, please revise the entire manuscript and standardize.

L190 – Please consider placing the text in the previous paragraph. 8-12 lines for each paragraph is suggested.

L194 – I believe it is table 2.

L191-198 – It is recommended to reorganize the results section in these lines. Particularly, the text previously and after tables and the format of the table content.

L292 – Please delete.

L306 - Please delete.

L311 – Please carefully revise the references format. For example, the ref 9 title is in uppercase (contrary to others), and DOIs are suggested (e.g. ref 1).

Please revise the English quality details.

Author Response

We sincerely thank the reviewer for the additional careful review and constructive comments on our revised manuscript. We have carefully addressed each suggestion as follows. All revisions in the manuscript are highlighted in red for clarity.

Comment 1 (L11–26):
Please double-check the journal template. Normally, the abstract does not have line spacing.

Response 1:
We have revised the Abstract section to follow the journal template, ensuring there is no unnecessary line spacing between lines.

Comment 2 (L89):
Please delete.

Response 2:
We have deleted the corresponding text at line 89 as suggested.

Comment 3 (L117):
Please delete.

Response 3:
The sentence at line 117 has been deleted as requested.

Comment 4 (L148):
Please revise the journal template and instructions for authors regarding formulas.

Response 4:
We reformatted the formula for the Scapular Index using the Equation Editor in Word, as recommended by the journal guidelines. The formula is now editable and follows the proper equation format (Equation 1).

Comment 5 (L175):
“p” not in italics in some lines, please revise the entire manuscript and standardize.

Response 5:
We have carefully checked the entire manuscript and standardized the formatting of all p values to italics consistently.

Comment 6 (L190):
Please consider placing the text in the previous paragraph. 8–12 lines for each paragraph is suggested.

Response 6:
We revised the Results section to merge shorter sentences where appropriate. Paragraph lengths were adjusted to remain within the suggested range (8–12 lines).

Comment 7 (L194):
I believe it is Table 2.

Response 7:
We corrected the numbering and references in the text to ensure that “Table 2” is properly cited in line 194.

Comment 8 (L191–198):
It is recommended to reorganize the results section in these lines. Particularly, the text previously and after tables and the format of the table content.

Response 8:
We reorganized the Results section to improve clarity. Specifically, we now provide a concise explanatory sentence immediately before each table (e.g., “The detailed values of muscle tone are presented in Table 2”), as recommended. This ensures that the narrative text and the tables are clearly connected, facilitating interpretation.

Comment 9 (L292):
Please delete.

Response 9:
The sentence at line 292 has been deleted as requested.

Comment 10 (L306):
Please delete.

Response 10:
The sentence at line 306 has been deleted as requested.

Comment 11 (L311):
Please carefully revise the references format. For example, the ref 9 title is in uppercase (contrary to others), and DOIs are suggested (e.g., ref 1).

Response 11:
We revised Reference 9 into sentence case, consistent with the other references:

  • Burcos, I.B.; Vătăman, A.V.; Onofrei, R.R.O. Myotonometric assessment of the short-term effects of manual therapy on upper trapezius myofascial trigger points. Timisoara Medical Journal 2024, 2023, 2, doi:10.35995/tmj20230202.

Additionally, DOIs were added where available (e.g., Ref 1 and others).

Comment 12 (English quality):
Please revise the English quality details.

Response 12:
We have had the entire manuscript professionally edited by a scientific English editing service (Editage). The grammar, clarity, and readability have been improved accordingly.

Final Statement

We sincerely appreciate the reviewer’s insightful comments, which have greatly improved our manuscript. We believe that the revised version now fully addresses the concerns raised.